# Hepatocellular Metabolic Abnormalities Induced by Long-Term Exposure to Novel Brominated Flame Retardant, Hexabromobenzene

**DOI:** 10.3390/toxics11020101

**Published:** 2023-01-21

**Authors:** Bohyun Shin, Se Hee Hong, Sumin Seo, Cho Hee Jeong, Jiyu Kim, Eunbin Bae, Donghee Lee, Jung Hoon Shin, Minki Shim, Sang Beom Han, Dong-Kyu Lee

**Affiliations:** College of Pharmacy, Chung-Ang University, 84 Heukseok-ro, Dongjak-gu, Seoul 06974, Republic of Korea

**Keywords:** novel brominated flame retardants, hexabromobenzene, metabolomics, gas chromatography–mass spectrometry, environmental pollution

## Abstract

Novel brominated flame retardants (NBFRs) are widely used to avoid environmental accumulation concerns and because of the regulations imposed on classical BFRs. However, recent studies have not revealed the negative effects of NBFR accumulation and exposure on humans. We conducted a metabolomics study on hexabromobenzene (HBB), one of the NBFRs, to investigate its effect on hepatocytes. Gas chromatography–mass spectrometry-based metabolite profiling was performed to observe metabolic perturbations by treating human livertissue-derived HepG2 cell lines with HBB for maximum 21 days. Metabolic pathway enrichment using 17 metabolite biomarkers determined via univariate and multivariate statistical analysis verified that long-term accumulation of HBB resulted in distinct diminution of eight amino acids and five other metabolites. Molecular docking of the biomarker-related enzymes revealed the potential molecular mechanism of hepatocellular response to HBB exposure, which disrupts the energy metabolism of hepatic cells. Collectively, this study may provide insights into the hidden toxicity of bioaccumulating HBB and unveil the risks associated with non-regulated NBFRs.

## 1. Introduction

Brominated flame retardants (BFRs), which suppress or delay combustion, are used in several industries including the production of electric and electronic equipment because of their outstanding efficacy and cost effectiveness [1]. BFRs that are frequently used in industries include polybrominated diphenyl ethers (PBDEs), hexabromocyclododecane (HBCD), and tetrabromobisphenol A [2].

Despite their versatility and cost effectiveness, some BFRs are environmentally toxic, durable, and bioaccumulative [1,2]. Therefore, the use, processing, and disposal of products containing BFRs leave residues that may be environmentally hazardous. These BFRs that accumulate in the environment (air, food, water, soil, etc.) can be exposed to the human body through inhalation and ingestion and are known to cause diabetes, neurobehavioral and developmental disorders, cancer, reproductive health effects, and changes in thyroid function [3,4,5]. Consequently, tetra-BDE, penta-BDE, HBCD, etc., have been designated as persistent organic pollutants (POPs) since 2009 and regulated by the Stockholm Convention [6].

As BFRs regulations tighten, novel brominated flame retardants (NBFRs) such as decabromodiphenyl ethane (DBDPE), 1,2-bis(2,4,6-tribromophenoxy)ethane (BTBPE) and hexabromobenzene (HBB), pentabromo-ethylbenzene (PBEB), and pentabromotoluene (PBT) have gradually replaced classical BFRs [7,8]. Although the chemical structures of NBFRs are different than those of traditional BFRs, most of their physicochemical features, including aromatic and lipophilic qualities, are comparable. [9]. According to previous NBFR environmental accumulation studies, NBFRs are also detected at high concentrations in the environment [9,10,11]. BTBPE, which is used as a substitute for octa-BDE, has been detected in house dust (1900 ng/g), sediments, and indoor air in the United States, China, and the United Kingdom. HBB has been detected at an average concentration of 8672 ng/g in sediments and up to 197 and 2451 ng/g in prawn and fish, respectively [12]. Other NBFRs, such as PBT and PBEB, have also been detected in the environment and found to accumulate in organisms such as fish, whales, and birds [9].

While there are many environmental accumulation studies on NBFRs, only a few studies reported the human risk of NBFRs exposure. According to a previous study, DBDPE can interfere with the hepatic metabolism and can even change the expression of cytochrome enzymes in rats [13]. Another study revealed that long-term exposure to DBDPE could affect thyroid function in humans [14]. Furthermore, American kestrels exposed to 2-ethylhexyl-2,3,4,5-tetrabromobenzoate (EHTBB) exhibited changes in thyroid function and oxidative stress [15]. NBFRs also affected neurobehavioral and reproductive development [16]. Among the major NBFRs used recently, we employed HBB to evaluate the toxicity. HBB is an emerging and widespread environmental pollutant, but only a few studies investigated its negative effect [17,18]. In a previous study, the exposure to HBB in the early life stages of zebrafish influenced developmental neurotoxicity by affecting central nervous system [19]. In addition, *Carassius auratus* exposed to HBB developed oxidative stress in liver and gill tissues and inhibited Na^+^, K^+^-ATPase activity [20]. These previous studies confirm the accumulation of HBB in different organisms and its adverse effects on their metabolism. However, the chronic toxicity of accumulating HBB and its exposure to humans remain severely underexplored.

Metabolomics-based studies on the toxicity of environmental pollutants provide a variety of metabolic evidences that are needed to understand the molecular mechanism of HBB. A metabolite, which is an end-product of sophisticated biochemical processes that maintain the homeostasis of cellular functions, can provide new insights into the cellular responses upon exposure to the pollutant. Given that the terminal biomolecular product reflects the distorted metabolism triggered by toxicant exposure, many studies elucidated the metabolic pathway relevant to the flame retardants [21,22]. Notably, the metabolomics approach can effectively demonstrate the subtle biochemical changes and the gradual disruption of homeostasis induced by these environmental pollutants, which do not show clear toxic reactions [23,24,25]. Combining metabolomics with in vitro experiments has the advantage of substituting animal testing and enabling a rapid identification of the toxicological properties of new compounds [26,27]. In particular, hepatic in vitro studies are widely conducted to evaluate the metabolic mechanisms related to toxicity [28] because the human hepatoma cell line maintains the activity of various enzymes that play an important role in activation and detoxification [29]. In a nutshell, in vitro metabolomics of hepatic cells have the potential to explain the effect of exposure to toxic substances [26,30].

In this study, we conducted in vitro metabolomics investigations to examine the toxicological effect of HBB. Since NBFRs are highly bioconcentrated in lipid deposits, we hypothesized that the influence of short-term and long-term exposure to HBB on the metabolic processes of hepatic cells is different. We employed gas chromatography–mass spectrometry (GC-MS)-based metabolic profiling with univariate and multivariate statistical analysis. Significant metabolic perturbations and the related pathways were discovered according to the short-term/long-term exposure group. The findings of this study demonstrate the hidden toxicity of bioaccumulating HBB.

## 2. Materials and Methods

### 2.1. Chemicals and Reagents

HBB was purchased from Tokyo Chemical Industry (Tokyo, Japan). Pyridine (99.5+%) and water (HPLC grade) and methanol (HPLC grade) were purchased from Thermo Fisher Scientific (Waltham, MA, USA). Chloroform (99.8%, HPLC grade) was purchased from DAEJUNG (Siheung, Republic of Korea), and dimethyl sulfoxide (DMSO, 99.9%, HPLC grade) was obtained from Samchun Chemical (Pyeongtaek-si, Republic of Korea). Thiazolyl blue tetrazolium bromide (98%), methoxyamine hydrochloride (98%), and N,O-bis(trimethylsilyl) trifluoroacetamide with trimethylchlorosilane (BSTFA 1% TMCS, 99%) were acquired from Sigma Aldrich (St Louis, MO, USA)

### 2.2. Cell Culture Conditions and Treatment with HBB

HepG2 cells were cultured in Dulbecco’s Modified Eagle Medium (4.5 g/L D-Glucose, L-Glutamine and 25 mM HEPES without sodium pyruvate, Gibco) containing 10% fetal bovine serum (FBS, Gibco) and 1% antibiotic-antimycotic (Gibco) under humidified air with 5% CO_2_, at 37 °C. In the HBB-exposure study, HepG2 cells were seeded in cell culture dishes (90 mm × 15 mm, SPL) with approximately 3 × 10^6^ cells/well and incubated with HBB for 0 to 21 days. For 21 days of exposure, cells were sub-cultured using a trypsin-EDTA solution in Dulbecco’s phosphate-buffered saline (PBS, Gibco) after reaching 80% confluence. HBB stock solution was dissolved in DMSO at a concentration of 1 mg/mL, and a final concentration of 1 µg/mL was obtained in the media by adding 0.1% (*v*/*v*) of the solution.

### 2.3. Cell Viability

HepG2 cells were exposed to HBB at various concentration of 0.05, 0.1, 0.5, and 1 µg/mL for 24 h and 48 h. HepG2 cells were seeded in a 96-well plate at a density of 1 × 10^4^ and 5 × 10^3^ cells/well, respectively, for 24 h and 48 h of exposure, respectively. Following the HBB exposure, cell medium was removed, and a thiazolyl blue tetrazolium bromide solution (5 mg/mL in PBS) was diluted to 0.5 mg/mL with FBS-free medium; next, to each well was added 200 µL of diluted reagent, and incubation was performed for 4 h at 37 °C and 5% CO_2_. After the incubation, medium was removed and dissolved with DMSO. Spectrophotometric absorbance analysis was performed at a wavelength of 570 nm using a plate reader (FlexStation 3, Molecular Device).

### 2.4. Metabolite Extraction and Derivatization

For the metabolomics study, extraction was performed after stabilization for 24 h. The cultured cells were rinsed with PBS and cold water. Next, cells were scrapped with 1 mL of cold water and transferred to a 1.5 mL Eppendorf tube, and cells were centrifuged to obtain a cell pellet, which was then stored at −80 °C until the next step.

To extract metabolites from the cell, a modified freeze–thaw extraction process was employed [31]. In this method, 700 µL of 50% cold methanol was added to the collected cell pellets, which were then subjected to repeated freeze/thaw in liquid nitrogen; this process was performed three times. After the freeze and thaw cycles, 350 µL of chloroform was added to precipitate the protein and perform a liquid–liquid extraction. Samples were vortexed for 10 s and centrifuged at 13,000× *g* for 5 min. The chloroform phase was then transferred to another tube, and the separated polar layer was filtered by a 0.45 µm polytetrafluoroethylene syringe filter (Whatman) and dried with nitrogen gas. Then, two-step derivatization reactions were performed for GC-MS analysis. In this case, 50 µL of methoxyamine hydrochloride solution (20 mg/mL in pyridine) was added to the dried metabolite tube, and methoxyamination was conducted for 30 min at 45 °C. Then, 50 µL of BSTFA 1% TMCS solution was used for trimethylsilyl derivatization, which was performed at 60 °C for 60 min. After the derivatization, the solution was centrifuged at 13,000× *g* for 3 min and subsequently transferred to a glass vial.

The protein layer was dried in a vacuum concentrator (micro-cenvac, N-BIOTEK), and then, bicinchoninic acid assay (BCA) analysis was performed using the Pierce™ BCA Protein Assay Kit (Thermo Fisher Scientific), and the vendor-provided protocol was used.

### 2.5. Data Processing and Statistical Analyses

The metabolites were identified by comparing their mass spectra with those available in the NIST08 library. For some metabolites that were difficult to identify, we compared the retention index of the peak shown by the derivatized standard compounds. GC-MS data were aligned using the MetAlign software (version 041012) and previously reported parameters [31]. Statistical analyses were performed using GraphPad Prism 7.0 and SIMCA 17 (Satorius), Metaboanalyst 5.0 (https://metaboanalyst.ca (accessed on 24 November 2022)). The metabolite data were normalized by BCA results and unit variance-scaled for multivariate analysis. Hotelling’s T^2^ analysis was used for the search of potential outliers, and consequently, these observations were excluded for further statistical data analysis. Partial least square–discriminant analysis (PLS-DA) and orthogonal partial least square–discriminant analysis (OPLS-DA) were secondly performed for the classification of the analyzed samples. *p*-value < 0.05 and variable importance for projection (VIP) > 1 were considered statistically significant. The univariate analysis of cell viability compared to control and metabolites alteration between short-term and long-term exposure used unpaired *t*-test. Pathway mapping was performed in VANTED (version 2.8.3) with the SBGN-ED add-on with an two-way ANOVA and Dunnett’s multiple comparisons test for each metabolite.

### 2.6. Molecular Docking

The protein structures were downloaded from the Protein Data Bank. The Protein Data Bank IDs for glutamic pyruvic transaminase (GPT), phenylalanine hydroxylase (PAH), ribokinase (RK), creatine amidinohydrolase (CAH), and branched-chain amino acids transferase 1 (BCAT) are 3IHJ, 1LRM, 2FV7, 3CTZ, and 7NTR, respectively. AutoDock4 and MGLTools (version 1.5.7) were used to generate the docked conformation between HBB and each protein [32]. The best binding conformation between the metabolite and the enzyme was selected out of 10 possible binding sites via a genetic algorithm.

## 3. Results

### 3.1. Cytotoxicity of Hepatic Cells under HBB Exposure

There was no morphological change associated with HBB exposure of cells (Appendix A). However, cell viability showed a significant difference. Cells were exposed to HBB in concentrations ranging from 0.05 to 1 µg/mL, and it was observed that the cell viability significantly decreased with the increasing HBB concentration (Appendix A). Cell viability exposed to 0.5 and 1 µg/mL of HBB exhibited distinctions of approximately 81% and 73% at 24 h exposure and 76% and 72% at 48 h exposure, respectively. At low doses (0.05 and 0.1 µg/mL), HBB did not show significant toxicity. Therefore, to better observe the HBB-induced toxicity, we selected a higher dose of HBB (1 µg/mL), which could lead to alteration of the metabolites.

### 3.2. Observation of Hepatic Metabolite Alterations Induced by HBB Exposure

Metabolite identification was performed based on metabolite profiling obtained using QC samples. Fifty-nine metabolites were obtained by comparing the mass spectra of every metabolite with mass spectral library. Sugars and some amino acids, whose mass spectra are identical to those of other metabolites in the same compound class, were confirmed by the retention index of the standard compound. Few metabolites having a coefficient of variation of over 30% within the QC samples (n = 5) were excluded as an outlier, and finally, a total of 37 metabolites were identified (Appendix A).

Principle component analysis (PCA) was performed based on 37 identified metabolites and 41 observations (control group n = 18, HBB-exposed group n = 18, and QC n = 5). The QC data were clustered at the center of the PCA score plots (Appendix A). The constructed PCA model was validated by evaluating the R^2^X and Q^2^ values (0.903 and 0.646, respectively). To discover remarkably altered metabolites between the control and HBB-exposed group, PLS-DA was employed, which is a supervised method (Figure 1A). Component 2 discriminated the samples into control and HBB-exposed groups. Metabolite signatures associated with HBB exposure were verified by considering both multivariate and univariate statistical parameters. In univariate analysis, 4 out of the 37 metabolites satisfied fold change (FC) > 1.2 or FC < 0.8 and *p*-value < 0.05 (Figure 1B). Eleven metabolites were confirmed with VIP > 1 in multivariate analysis (Figure 1C). Three metabolites, namely urea, myo-inositol 1-phosphate, and glucose, satisfied both univariate and multivariate analysis criteria. As a result, compared to the control group, a total of three metabolites were confirmed as the key metabolites of the HBB-exposed group (Figure 1D and Appendix A).

### 3.3. Metabolic Perturbation Analysis of Acute and Chronic Toxicity Induced by HBB Exposure

To examine the perturbation between the acute and chronic metabolites, the changes in the metabolites on each day of the HBB exposure were observed. Firstly, an outlier was verified using PCA and Hotelling’s T^2^ ellipse 5% plot. As a result, the group exposed to HBB for 10 days was identified as an outlier and excluded. As shown in the Figure 2A, PLS-DA score plots with each day of HBB exposure are clustered, showing the short-term (3 and 7 days) and long-term (15, 17, and 21 days) features. OPLS-DA was performed by dividing the samples into long-term and short-term groups to maximize the discrimination and to identify the metabolite biomarkers between the groups. The short-term and long-term groups are distinctly discriminated from each other in OPLS-DA score plots (Appendix A). Univariate and multivariate statistical analyses were performed to identify the key metabolites between the groups. A total of 17 metabolites, with FC > 1.2 or FC < 0.8, *p*-value < 0.05, and VIP > 1, were distinguished from short-term and long-term exposure groups (Table 1); specifically, we found eight amino acids, three sugars, one organic acid, and five other compounds. For all the metabolites, the relative abundance of short-/long-term exposure was compared using a box plot (Figure 2B). Most of the metabolites declined upon prolonged exposure, but the concentration of d-erythrotetrofuranose increased (1.36 fold).

### 3.4. Pathway Enrichment Analysis Affected by HBB Exposure

Pathway enrichment analysis with the metabolite biomarkers was employed to identify the significantly distinct metabolic pathways between acute and chronic toxicity (Figure 3 and Appendix A). As a result of pathway enrichment, 12 metabolic pathways were found to be related with chronic toxicity. *p*-value < 0.05 indicated that long-term exposure to HBB significantly affected the valine, leucine, and isoleucine biosynthesis; arginine biosynthesis; fructose and mannose metabolism; and pantothenate and CoA biosynthesis. Particularly, in the valine, leucine, and isoleucine biosynthesis pathway, three out of the eight metabolites in the pathway were detected with the lowest *p*-value; in this case, three metabolites (valine, leucine, and isoleucine) tended to decrease during the long-term exposure. Arginine biosynthesis, which included 3 out of the 14 metabolites (aspartic acid, urea, and glutamate), was also affected by long-term HBB exposure. In the phenylalanine, tyrosine, and tryptophan biosynthesis, only one metabolite exhibited a significant alteration out of the four metabolites in the pathway. The last three pathways, which are listed at the bottom of Appendix A, were found to be of low importance (as only two metabolites out of the >30 metabolites) in HBB exposure.

Metabolic pathway mapping was performed by using 35 metabolites and their related metabolic pathways (Figure 4). We excluded aminomalonic acid and d-erythrotetrofuranose from the mapping because they are not closely located in the metabolomic network of the other 35 metabolites. The metabolomic alterations were interpreted in the pathway map based on the determined effect of pathway enrichment on the short-/long-term exposure. Metabolites included in the valine, leucine, and isoleucine biosynthesis pathway, which were the most significant metabolic mechanisms in chronic toxicity, were significantly down-regulated over time. Valine was significantly different from the control group on the 15, 17, and 21 days of HBB exposure; isoleucine was significant at 17 and 21 days; and leucine was only significant at 21 days. In addition, urea, aspartic acid, and glutamate, which are in the arginine biosynthesis pathway, were also significantly down-regulated and time-dependent. In common, all three metabolites showed no significant difference in long-term HBB exposure. Metabolites related to fructose and mannose metabolism, such as fructose, mannose, and fructose 1-phosphate (F1P), were reduced by long-term exposure to HBB. Further, three amino acids (serine, threonine, and phenylalanine) were distinctly decreased after 17 and 21 days of exposure. Metabolites involved in the TCA cycle did not show significant metabolic changes except for malate.

### 3.5. Molecular Docking

Molecular docking was performed to verify the potential target enzyme of the toxic HBB ligand further. We simulated a molecular docking of HBB with five enzymes that contribute to the top five significant metabolites in short-term/long-term comparisons (Table 1). Docking results were based on binding energy (ΔG_bind_) and inhibition constant (Ki), wherein smaller ΔG_bind_ and Ki values indicated the stable HBB–enzyme bindings. We measured the ΔG_bind_ of GPT, PAH, RK, CAH, and BCAT, whose inhibition may change the expression of alanine, phenlyalanine, ribose, urea, and valine, respectively. The ΔG_bind_ values predicted by the molecular docking were −5.76, −6.92, −6.52, −5.60, and −6.30 kcal/mol for GPT-HBB, PAH-HBB, RK-HBB, CAH-HBB, and BCAT-HBB, respectively. The results of each run are presented in Appendix A. Three-dimensional (3D) molecular docking structures between HBB and each protein are shown in Figure 5.

## 4. Discussion

We investigated the metabolic responses to HBB, a bioaccumulative NBFR, on human hepatic cells using a metabolomics approach. We monitored the toxic effects and metabolite alterations caused by HBB exposure. In particular, the perturbation of metabolites and the metabolic pathway were affected by the long-term exposure to HBB. This result evidences the disruption of homeostasis due to the chronic toxicity of HBB or other NBFRs, which bioaccumulate because of their lipophilic properties.

We verified that compared to short-term exposure, long-term exposure to HBB results in obvious changes in the metabolic mechanisms. Most of the metabolites in the control and short-term exposed groups exhibited similar concentrations, and this similarity resulted in a relatively unclear clustering between control and HBB-exposed groups (Figure 1A) in PLS-DA score plots. By contrast, the short-term and long-term groups were clearly separated in the time-dependent classification (Figure 2A) and OPLS-DA score plots (Appendix A). The metabolic pathway networks, with a bar plot of 35 metabolites, even indicate that the concentrations of most of the metabolites significantly decreased within 15 to 17 days of HBB exposure (Figure 4). Eight amino acids (valine, glutamate, leucine, serine, threonine, isoleucine, phenylalanine, and aspartate), lactate, fructose, F1P, malate, and urea are metabolite signatures whose concentrations decreased noticeably only in response to chronic toxicity (15 to 21 days of exposure). In contrast, the concentrations of upstream metabolites related to the central carbon metabolism, including glucose, ribose, and 3-phosphoglyceric acid, rapidly decreased upon HBB exposure. This result implies that the acute and chronic toxicity triggers the impaired homeostasis by different molecular mechanisms. Together with the environmental persistency of HBB, the industrial use of this NBFR results in toxicity issues that cause gradual loss of cellular functions.

Long-term exposure to HBB has been shown to decrease the level of metabolites, which play an important role in maintaining the metabolic function. Branched-chain amino acids (valine, leucine, and isoleucine), which are related to protein and neurotransmitter synthesis, down-regulate upon prolonged exposure to HBB. BCAAs activate the mammalian target of rapamycin (mTOR) and also stimulate the nuclear import of polypyrimidine–tract binding protein, which binds to albumin mRNA and increases its translation in HepG2 cells [33,34]. Low concentrations of BCAAs are related to suppression in albumin synthesis and its decreased secretion as well as chronic liver disease [35,36]. In addition, urea showed a significant decrease in the HBB-exposed and chronic toxicity groups, which could indicate HBB triggered an abnormal urea cycle activity for processing excess nitrogen mainly generated by ammonia within the liver [37]. Accumulation of ammonia induces oxidative/nitrosative stress in astrocytes and is known to cause neurotoxicity [38]. In addition, studies have shown that the accumulation of citrulline and arginine, which are metabolites related to the urea cycle, can damage the cellular antioxidant response [39]. Observing changes in the expression of enzymes or genes (carbamoyl phosphate synthetase I, CPS-1; ornithine transcarbamylase, OTC; argininosuccinate synthetase, ASS; argininosuccinate lyase, ASL; arginase 1, ARG1) related to the urea cycle in vitro or in vivo can further solidify the metabolomic results selected as biomarkers in the paper. However, our results were performed in vitro, and it will be necessary to verify in future studies whether this results in whole-body metabolome changes following actual HBB exposure. Further, the concentrations of phenylalanine and inosine are found to diminish with prolonged HBB exposure. Phenylalanine is an important amino acid because its hydroxylation produces tyrosine, which acts as a precursor to several metabolites such as epinephrine [40]. Inosine is a central intermediate in purine biosynthesis and degradation pathways and is also involved in neuronal signaling [41]. According to a previous study, metabolic changes in the brains of mice exposed to BDE-47 for 30 days affects the purine, phenylalanine, and tyrosine metabolisms, which may be related to the pathology of Parkinson’s disease [42]. The reduced concentration of phenylalanine is also consistent with the results of our HBB exposure study. Collectively, these results suggest that HBB causes hepatotoxicity and chronic toxicity, and further studies are needed to understand the toxic mechanism in humans. Notably, liver function enzymes, aspartate aminotransferase (AST), and alanine aminotransferase (ALT), which are used as markers to evaluate liver function, could be closely related to HBB exposure. When the liver is damaged, intracellular ALT and AST enter the bloodstream, increasing serum ALT and AST levels and decreasing ALT and AST levels in liver tissue [43]. According to a study conducted in relation to flame retardants, the serum concentration of AST increased in ICR mice exposed to TBBPA at a high dose (1400 mg/kg body weight) [44], and the concentrations of AST and ALT decreased in liver tissues exposed to 2-ethylhexyl diphenyl phosphate [45]. Given these associations, exposure to HBB could have a potential effect on liver function, which should be investigated in future in vitro or in vivo study.

Molecular docking revealed the potential inhibition of biomarker-related enzymes [46]. GPT, PAH, RK, CAH, and BCAT relate the metabolism of the five metabolites alanine, phenylalanine, ribose, urea, and valine, respectively; these metabolites significantly altered between acute/chronic toxicity. Out of these enzymes, BCAT, which catalyzes the 2-oxosiovalerate to valine reaction, exhibits strong binding (ΔG_bind_: −6.3 kcal/mol) with HBB. This strong HBB binding could induce a subsequent decrease in valine concentration as a mechanism of chronic toxicity. Based on the 3D molecular docking structure and the binding energy, we hypothesize that HBB may have a potential inhibitory activity to BCAT. The other four enzymes (GPT, PAH, RK, and CAH) also exhibit relatively low binding energies, and thus, it is believed that they could potentially affect the enzymes. It is presumed that the HBB is docked to the enzyme as a ligand and acts as an inhibitor, thereby inhibiting the activity of the enzyme, and consequently, the production of metabolites in the reaction decreases. As our molecular docking results indicate potential HBB inhibition to biomarker-related enzymes without actual data, further studies on this point should be conducted in future study.

In summary, we found that the altered patterns of the metabolome of HepG2 were distinct for the short-term and long-term exposure groups. In addition, we demonstrated the adverse effects of HBB accumulation on the metabolic pathway. Further, we elucidated the underlying cytotoxic mechanism of HBB using the molecular docking data on the potential binding of the HBB. The findings of this study aid in the understanding of the biochemical mechanisms of hepatic response to HBB and provide a new route for future biochemical studies on NBFRs. Such studies are crucial to understand the risks associated with the environmental accumulation of NBFRs.

## 5. Conclusions

In this study, we discovered that accumulation of HBB, one of the NBFRs, may produce metabolic abnormalities in human hepatocytes. A metabolic mechanism caused by long-term exposure of HBB mostly relates to the amino acids or energy metabolism, which is distinct from those induced in the short-term or no-exposure groups. Using molecular docking, we estimated the potential underlying HBB affinity for the biomarker-related enzymes. These results provide new insight into the harmful processes of HBB against hepatocellular metabolism. It could further propose a new approach for toxicological assessment of bioaccumulating NBFRs to fully comprehend the risks associated with their environmental accumulation.

## Figures and Tables

**Figure 1 toxics-11-00101-f001:**
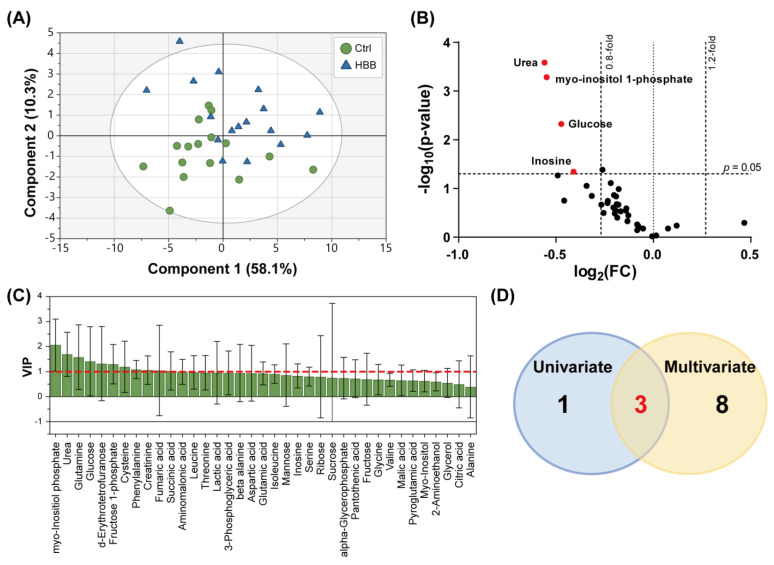
Univariate and multivariate statistical analysis of the metabolic data. (**A**) PLS-DA score plots of the control (green circle) and HBB-exposed groups (blue triangle). (**B**) Volcano plot of 37 metabolites. The *x*-axis corresponds to log_2_ (fold change) and *y*-axis to −log_10_ (*p*-value) values, respectively. Metabolites with *p*-value < 0.05 and FC > 1.2 or FC < 0.8 are marked with red. (**C**) VIP value and (**D**) Venn diagram obtained from the univariate and multivariate statistical analyses.

**Figure 2 toxics-11-00101-f002:**
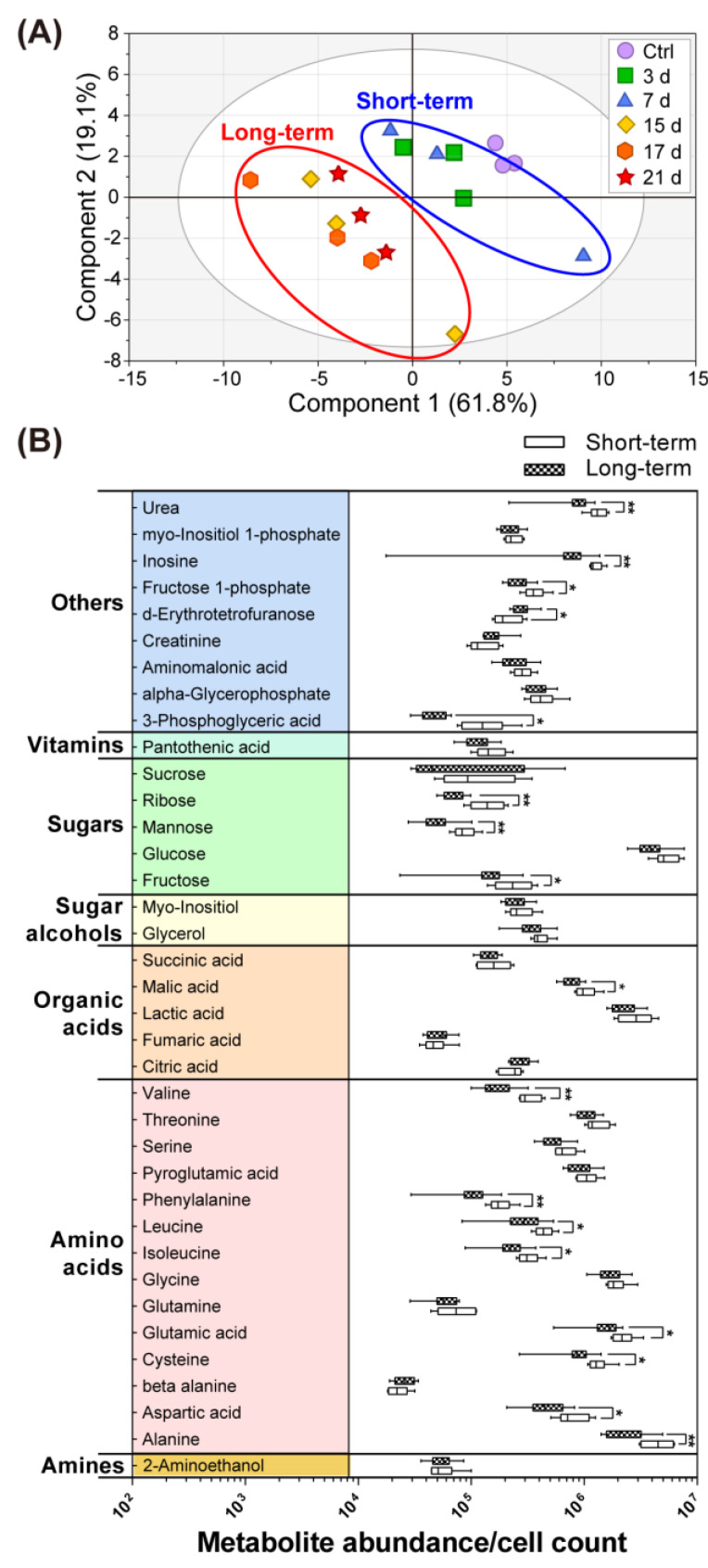
Observation of metabolic changes according to short-/long-term exposure. (**A**) PLS-DA score plots of the metabolites exposed to HBB for different number of day. (**B**) Box plots of short-term and long-term groups with metabolite abundance normalized by cell count. The unit of *x*-axis represents the normalized abundance data on a log scale. Significant differences are indicated by *t*-test results (* *p* < 0.05, ** *p* < 0.01).

**Figure 3 toxics-11-00101-f003:**
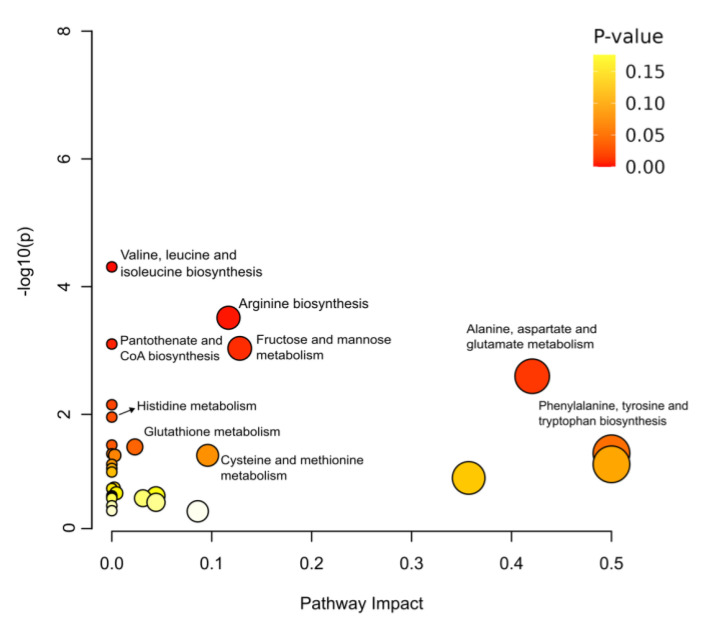
Metabolite pathway enrichment analysis affected by short-term and long-term exposures. Each circle represents a metabolic pathway, and its size indicates the enrichment ratio (the impact of pathway); the colors demonstrate the corresponding *p*-values (red, lowest value; yellow, highest value).

**Figure 4 toxics-11-00101-f004:**
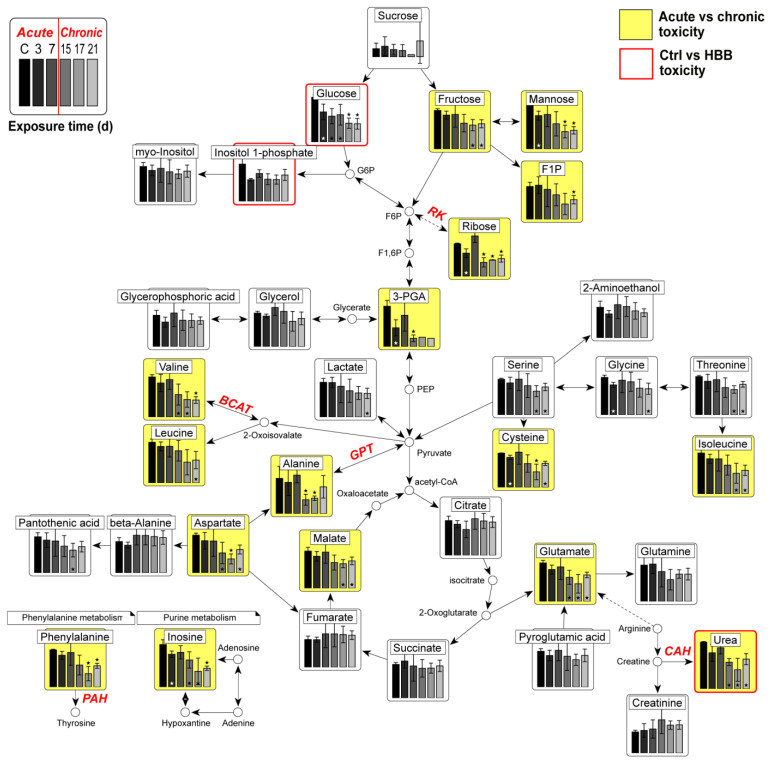
Metabolic pathway networks with a bar plot of 35 metabolites. The proposed metabolic pathway is presented based on the KEGG database. The *x*-axis represents the control group and HBB exposure days, and the *y*-axis represents the normalized abundance. Significant differences are indicated by *t*-test results (* *p* < 0.05). Red borders indicate significant metabolites comparison of control/HBB-exposed groups. Yellow boxes indicate the metabolites that are significant in the acute versus chronic statistical analysis. Five enzymes, i.e., GPT, PAH, RK, CAH, and BCAT, which are used for molecular docking, are indicated on the pathway (Abbreviations: 3-PGA, 3-phosphoglyceric acid; F1P, fructose 1-phosphate; PEP, phosphoenolpyruvate; F1,6P, fructose 1,6-bisphosphate; F6P, fructose 6-phosphate; G6P, glucose 6-phoshpate).

**Figure 5 toxics-11-00101-f005:**
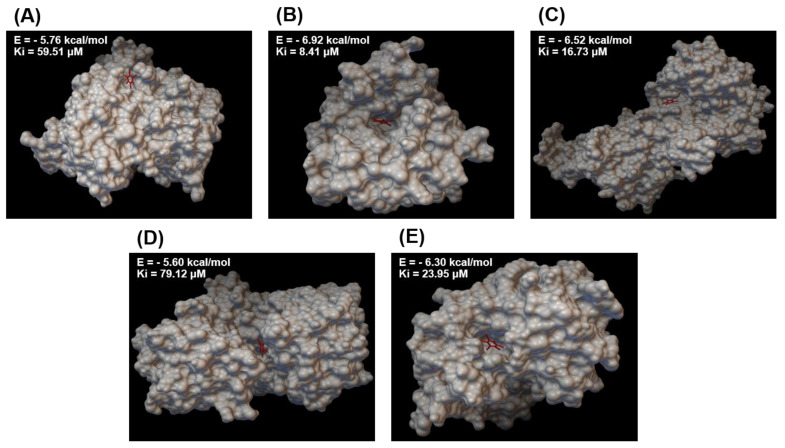
Overlay of HBB on five metabolite biomarker-related enzymes. HBB (red stick) is shown superimposed on a surface drawing of (**A**) GPT, (**B**) PAH, (**C**) RK, (**D**) CAH, and (**E**) BCAT. E, binding energy (kcal/mol); Ki, inhibition constant (µM).

**Table 1 toxics-11-00101-t001:** Statistical results of comparison of short-term and long-term HBB exposure group.

No.	Metabolites	VIP Value	*p*-Value	Fold Change
1	Valine	1.258	0.0012	0.5314
2	Ribose	1.363	0.0020	0.5026
3	Phenylalanine	1.266	0.0045	0.5692
4	Alanine	1.188	0.0063	0.5354
5	Urea	1.254	0.0063	0.6518
6	Mannose	1.183	0.0073	0.6026
7	Inosine	1.176	0.0076	0.6079
8	Cysteine	1.217	0.0140	0.6463
9	Aspartic acid	1.142	0.0175	0.5953
10	3-Phosphoglyceric acid	1.254	0.0225	0.3386
11	Glutamic acid	1.181	0.0239	0.6844
12	Fructose	1.106	0.0240	0.7314
13	Malic acid	1.090	0.0272	0.7595
14	Isoleucine	1.095	0.0293	0.6942
15	d-Erythrotetrofuranose	1.093	0.0308	1.3647
16	Leucine	1.038	0.0398	0.6883
17	Fructose 1-phosphate	1.051	0.0435	0.6184

## Data Availability

All data are provided in paper and Appendix A.

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
