# Peer review of "Hepatocellular Metabolic Abnormalities Induced by Long-Term Exposure to Novel Brominated Flame Retardant, Hexabromobenzene"

_toxics, 2023, doi:10.3390/toxics11020101_

Round 1

Reviewer 1 Report

Review Reports: Accept after Minor Revisions

Hepatocellular metabolic abnormalities induced by long-term exposure of novel brominated flame retardant, hexabromobenzene

I find this work very interesting and deals with a subject of public health; it is organized in a logical and clear manner. The literature review used contains relevant elements that give more value to this work.

The work realized by Bohyun Shin aimed to conduct in vitro metabolomics investigations to examine the toxicological effect of hexabromobenzene (HBB), one of the NBFRs that are widely used to avoid accumulation problems in the environment, in order to study its effects on hepatocytes.

Bohyun Shin, therefore, found through the study of Cytotoxicity of hepatic Cells, Molecular docking, Observation of Hepatic Metabolite Alterations, Metabolic Perturbation Analysis of Acute and Chronic Toxicity, and Pathway enrichment analysis affected by HBB exposure, that the accumulation of HBB, can produce metabolic abnormalities in human hepatocytes. These results provide new insight into harmful processes of HBB to hepatocellular metabolism. It could further propose a new approach for toxicological assessment of bioaccumulating NBFRs to fully comprehend the risks associated with their environmental accumulation.

The paper is generally well written and structured, the manuscript clear, relevant for the field, and presented in a well-structured manner, the quality of the figures doesn’t need to be reviewed, the methods are adequately described and the results provide an advancement of the current knowledge. However, in my opinion, the paper has some shortcomings in regard to some parts.

Below I have provided some remarks on the text:  

-      The title could be replaced by: Hepatocellular metabolic abnormalities induced by long-term exposure to novel brominated flame retardant, hexabromobenzene

-      The references mentioned are very pertinent but mostly old (46,6% of the references are recent).

Reviewer 2 Report

The Shin et al., 2022, manuscript ID 2133692 addresses the effect of long-term exposure of novel hazardous chemical hexabromobenzene in the hepatic toxicity. A search on Pubmed.gov for the terms "Liver" and " hexabromobenzene " keywords resulted in 29 studies on liver toxicity but only 3 studies on searching "Liver" and " hexabromobenzene" and "metabolite". The manuscript is well written with quality figures and has potential for publication.

There are few queries and few suggestions which makes this manuscript more representable to be publish.

1.      It will be great if the authors can show the expression of various enzymes involved in urea cycle? Authors must include those results?

2.      Did the authors have check the initial activity of ALT AST enzymes in the liver?

3.      Did the authors check levels of ammonia and urea in the liver of rat treated with HBB?

4.      Futher the authors showed check the histomorphology of the liver of HBB treated animals> 

Reviewer 3 Report

The authors have conducted an interestingly and timely research about hepatocellular metabolic abnormalities induced by long-term exposure of novel brominated flame retardant, hexabromobenzene.

Minor recommendation:

2. Materials and Methods, should include at Statistical analyses the p-value that was considered statistically significant should be inserted;

Are there any limitations to this study? If so, this aspect should be included, in order to help future authors and their studies.

Round 2

Reviewer 2 Report

Authors have tried to justify the concern raised by me. But some parts must need experimental justifications. 

Reviewer 3 Report

Dear Authors,

I appreciate your interest in the suggestions made, and I hope I've helped improve your article.